# Framework for Functional Clustering and Source-to-Sensor Reconstruction of Temporally Evoked Neural Features

* Equal Contribution

Adarsh Mukesh *
*Hearing Sciences*
*Purdue University*
USA

Shourya Verma *
*Computer Science*
*Purdue University*
USA

Varsha Mysore Athreya
*Otolaryngology*
*Stanford University*
USA

Ananth Grama
*Computer Science*
*Purdue University*
USA

Michael G. Heinz
*Hearing Sciences*
*Purdue University*
USA

*Abstract*—Noninvasive neural recording methods like electroencephalography (EEG) offer high temporal resolution for capturing neural activity. However, interpreting EEG data is challenging scalp-recorded signals (sensor space) reflect complex, integrated activity from multiple cortical regions (source space), complicating the reconstruction of underlying neural dynamics. Traditional approaches like minimum norm estimation require extensive subject-specific data, including MRI scans, precise electrode placement, and detailed anatomical atlases. To address these limitations, we propose a two-part framework: (1) an unsupervised biLSTM autoencoder that reveals clustering patterns in EEG electrode activations and their temporal dynamics during auditory stimulus processing; and (2) a deep learning architecture to predict temporally evoked neural features in sensor space EEG from source representations using a dual-path network with independent stimulus processing and dilated convolutional layers.

The clustering identifies evolving spatiotemporal co-activation patterns between stimulus onset and gaps, revealing functional reorganization. The reconstruction network reduces input dimensionality and integrates features via convolutional blocks with residual connections, trained using a hybrid loss that combines feature-based and spectral terms. Our results demonstrate accurate reconstruction of stimulus-related neural correlates and reveal topographical patterns consistent with the clustering findings. The model generalizes well across subjects. By analyzing both functional organization in sensor signals and source-to-sensor mappings, our framework enhances understanding of EEG transformations. This has significant implications for brain-computer interfaces, neuroimaging, and EEG processing where accurate reconstruction and interpretation are essential.

*Index Terms*—EEG; Dimensionality Reduction; Autoencoder; CNN; Reconstruction; Brain Computer Interface

## I. INTRODUCTION

Electroencephalography (EEG) provides flexible, low-cost neural activity measurement with high temporal resolution, but mapping cortical sources to scalp recordings remains challenging due to anatomical variability and standardization difficulties across subjects. We develop a deep learning model that directly learns source-to-sensor mappings from data, accounting for inter-subject and experimental variability. Understanding cortical-to-scalp EEG relationships is crucial for brain-computer interfaces, neurological diagnostics, and

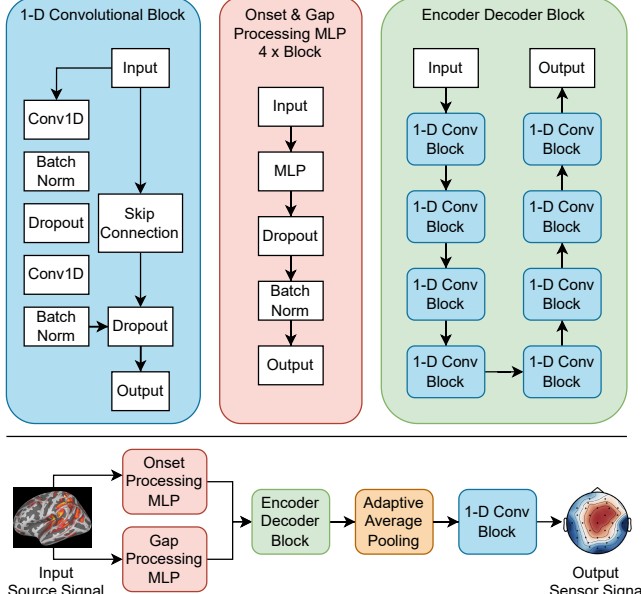

Fig. 1. DualPathEEG source-to-sensor reconstruction network autoencoder (AE) architecture diagram

cognitive neuroscience. While EEG's accessibility makes it widely used, experimental variability (cap placement, anatomical differences) degrades signal quality. Linear inverse methods like minimum norm estimates (MNE) [1] and beamformers fail to capture nonlinear source-sensor dynamics, losing spatiotemporal information. Numerical EEG forward solutions have been extensively studied. Grech, Cassar, Muscat, *et al.* [2] emphasize accurate head models using Finite Element (FEM) and Boundary Element Methods (BEM) for optimal accuracy-efficiency balance. Wolters, Anwander, Tricoche, *et al.* [3] examine conductivity assumption effects. Hybrid approaches include Erdbrügger, Westhoff, Höltershinken, *et al.* [4]'s CutFEM, combining hexahedral and tetrahedral meshing for improved computational efficiency.

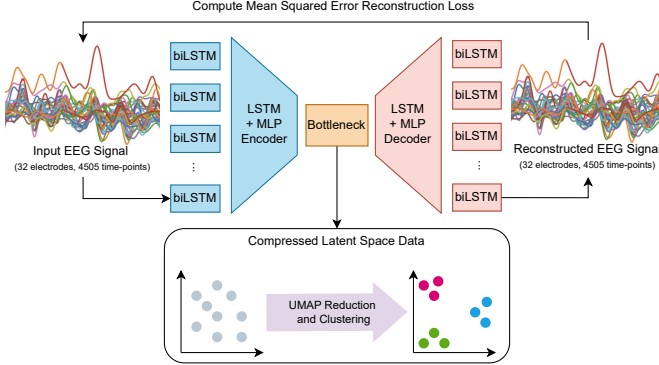

Fig. 2. BiLSTM autoencoder (AE) model along with UMAP reduction and clustering architecture diagram

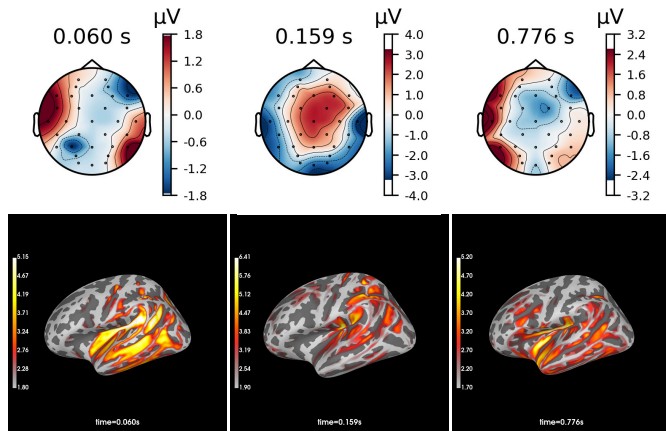

Fig. 3. Sample sensor space EEG topographical maps (top) and their corresponding source activation maps (bottom)

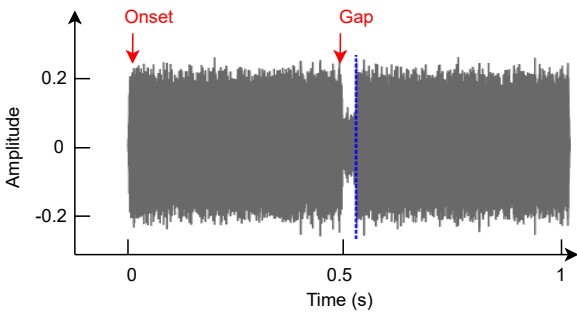

Fig. 4. Auditory stimulus structure used in the EEG study

Deep learning provides efficient forward modeling alternatives. Sun and Sclabassi [5] demonstrated neural networks approximating forward solutions faster than traditional methods. Xiong, Ma, and Li [6] proposed autoencoders for EEG denoising, improving time and frequency domain metrics. Cisotto, Zancanaro, Zoppis, *et al.* [7] developed variational autoencoders for robust multi-channel reconstruction. Physics-informed approaches include Wei, Lou, Wang, *et al.* [8]'s sparse basis networks with Gaussian source priors and Panwar, Rad, Jung, *et al.* [9]'s Wasserstein GANs for realistic EEG simulation. Cross-subject and cross-stimulus generalization remains challenging. Full time-series reconstruction models suffer from experimental factor variability, while GANs suit image data better and rely on synthetic datasets. Our Dual-PathEEG reconstructs 32-channel sensor EEG from 20,484-channel source data, addressing distribution scaling and patient variability challenges including cap placement and noise. Focusing on behaviorally relevant events rather than full time-series reduces complexity while preserving critical information, enabling real-time brain-computer interface applications. We propose a bi-directional LSTM autoencoder [10] clustering 32-channel EEG responses to reveal auditory event patterns. Our DualPathEEG architecture integrates convolutional neural network (CNN) blocks [11] to reconstruct source-to-sensor signal changes, capturing local and global dependencies for enhanced signal quality and interpretability.

## II. METHODS

### A. Auditory Stimulus EEG Dataset

We used a lab-generated EEG dataset (n=38 adults) recorded with tone-gap auditory stimuli consisting of two 4kHz pure tones (0.5s each) separated by gaps of 16ms, 32ms, or 64ms, presented with 0.5s inter-stimulus intervals and embedded in octave band noise centered at 4kHz. EEG was recorded using a 32-electrode BioSemi system at 4096 Hz sampling rate. Preprocessing included re-referencing to average earlobe electrodes, 1-40 Hz band-pass filtering, artifact removal (ocular via signal-space projections, muscular via 150$\mu$V peak-to-peak threshold rejection), and epoching from

300ms pre-stimulus to second tone offset with 200ms pre-stimulus baseline correction. Source reconstruction employed averaged evoked responses (20 trials per average) using MNE-Python's fsaverage template with corresponding source (src) and boundary element method (BEM) models. Forward modeling used the fsaverage BEM for realistic head conductivity, while the cortical surface source space constrained estimates to anatomically plausible locations. After electrode-anatomy alignment and noise covariance estimation, minimum norm estimation computed the inverse model to project sensor data to cortical activation maps (Figure 3). The auditory paradigm (Figure 4) targets two key events: stimulus onset, which activates primary auditory filters marking environmental change, and gap detection, where neural adaptation to continuous stimulation creates increased activity following perturbations. This reflects mismatch negativity at larger scales [12] and deviant detection at smaller scales [13], phenomena observed across species [14]. Figure 5 shows processed 32-electrode sensor signals and onset/gap value distributions across 20 patients for both sensor and source datasets.

### B. Unsupervised Clustering Analysis

To uncover underlying patterns in neural responses during onset and gap stimulus events, we design a deep learning-based feature extraction model. The framework consists of

a bi-directional LSTM autoencoder [10] for temporal feature learning with Uniform Manifold Approximation and Projection (UMAP) [15] dimensionality reduction for visualization and clustering analysis. Our bi-directional LSTM autoencoder employs an encoder-decoder structure where the encoder $f_{\text{enc}}$ maps input sequence $X$ to latent representation $Z$, and decoder $f_{\text{dec}}$ reconstructs the input: $Z = f_{enc}(X)$ for $Z \in \mathbb{R}^L$ and $\hat{X} = f_{dec}(Z)$ for $X \in \mathbb{R}^{T \times C}$, where $T$ represents the temporal dimension (sequence length) corresponding to time samples in each EEG epoch, $C$ is the electrode count (32), and $L$ is the latent space dimension (64) representing compressed input signal representation. The encoder pathway incorporates an input projection layer reducing sequence length through linear transformation with layer normalization and ReLU activation. Figure 2 illustrates the complete biLSTM autoencoder architecture and processing flow with UMAP reduction and clustering.

Learned latent representations undergo UMAP processing to create two-dimensional embeddings configured with $n_{\text{neighbors}} = 4$ and min_dist $= 0.1$ parameters, capturing local electrode channel relationships while preserving global structure. Using UMAP embeddings, we implement co-occurrence based clustering (Figure 6). The co-occurrence matrix $C$ captures frequency of similar activation patterns between electrode channels:

$$C_{ij} = \frac{1}{N} \sum_{k=1}^{N} \mathbf{1}(d_{ij}^k \leq \tau), \tag{1}$$

where $N$ represents total recordings, $d_{ij}^k$ is the Euclidean distance between electrode channels $i$ and $j$ in recording $k$, and $\tau$ is the distance threshold (1.5). Stable clusters are identified using DBSCAN clustering [16] with parameters $\epsilon = 0.5$ and min_samples $= 3$, evaluated using Silhouette score averaging $\approx 0.7$. Clustering analysis was performed on two time segments: 300ms before and after the gap event, enabling temporal pattern comparison across different phases of auditory processing.

### C. Source-to-Sensor Reconstruction

The DualPathEEG architecture reconstructs sensor space EEG from source space representations by focusing on neural correlates of two temporal events (stimulus onset and gap) rather than full time-course reconstruction. For each event, normalized RMS quantifies signal strength change: $RMS_{\text{norm}} = RMS_{\text{Post}}/RMS_{\text{Pre}}$ using 300ms post-event windows. The task transforms high-dimensional source signals (batch-size, 20484, 2) to sensor space (batch-size, 32, 2), where each sample contains RMS change indices for onset and gap events. The architecture employs dual processing pathways. The first pathway implements independent stimulus processing through parallel multilayer perceptrons (MLPs) [17] performing two-stage dimensionality reduction: 20,484 $\rightarrow$ 8,192 $\rightarrow$ 4,096 $\rightarrow$ 1,024 features via transformations with batch normalization, ReLU activation, and 0.1 dropout regularization. The second pathway uses eight CNN blocks in encoder-decoder architecture with dilated convolutions at

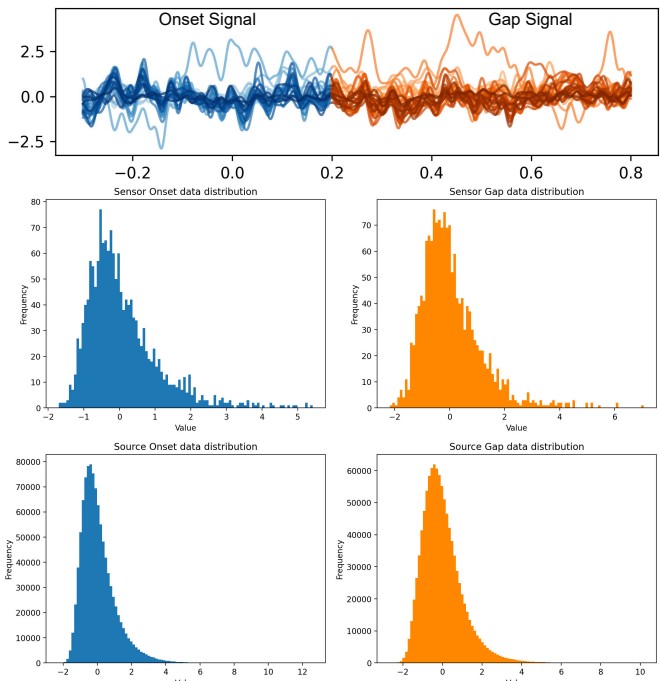

Fig. 5. Example of full EEG signal (top) along with sensor and source data distribution (bottom) of calculated RMS values for onset and gap for a batch of 20 samples.

progressively increasing rates (1, 2, 4, 8) for encoding and decreasing rates (4, 2, 1, 1) for decoding. Each CNN block incorporates dual Conv1D layers (kernel size 5), batch normalization, ReLU activation, dropout, and skip connections [18] for gradient flow and feature preservation. Multi-scale feature capture uses three parallel adaptive average pooling operations (128, 64, 32 temporal resolutions), fused through Conv1D with batch normalization, ReLU, and dropout. Final mapping to output dimensions uses a Conv1D layer, effectively capturing temporal dynamics and frequency characteristics while maintaining computational efficiency. Training employs a hybrid loss function combining feature and spectral components:

$$\mathcal{L}_{\text{feature}} = 0.5 \cdot \mathcal{L}_{\text{MSE}} + 0.5 \cdot \mathcal{L}_{\text{SmoothL1}}$$
$$\mathcal{L}_{\text{spectral}} = \frac{1}{N} \sum_{i=1}^{N} (|\mathcal{F}(y_i)| - |\mathcal{F}(\hat{y}_i)|)^2 \tag{2}$$
$$\mathcal{L}_{\text{total}} = \alpha \mathcal{L}_{\text{feature}} + \beta \mathcal{L}_{\text{spectral}}$$

Where $\mathcal{F}$ is fourier transform, and $\alpha = 0.7$, $\beta = 0.3$, ensuring both time-domain feature accuracy and spectral fidelity. Stimulus-wise normalization uses dataset-wide statistics with independent onset/gap parameters. The 80/20 train/validation split across patients prevents data leakage. Training uses AdamW optimizer [19] (learning rate 1e-4, weight decay 1e-5) with adaptive scheduling, gradient clipping (max norm 0.5), 200 epochs, batch size 32, and 5-fold cross-validation.

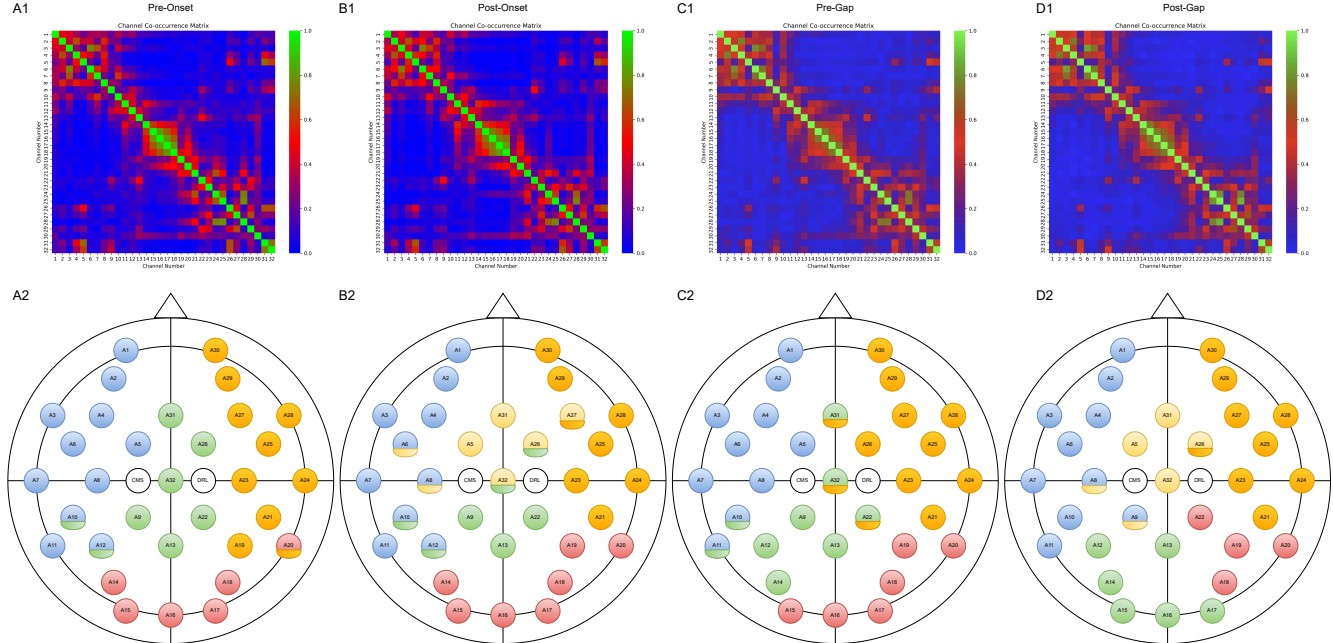

Fig. 6. Cluster co-occurrence matrix for 32 electrode sensor data with their respective stable clusters mapped on the electrode montage. Figures A and B, represent pre- and post onset signal. Figures C and D represent the pre- and post gap signal.

## III. RESULTS

### A. Cluster Analysis Reveals Patterns In Stimulus Temporal Dynamics

Unsupervised clustering was performed on 300ms evoked neural signals before and after gap occurrence using the biLSTM autoencoder architecture. Our analysis revealed distinct spatiotemporal patterns of electrode channel co-activation across different temporal segments as shown in Figure 6. Figure 6 A1-D1 displays electrode channel co-occurrence matrices for pre- and post-onset and gap periods, while Figure 6 A2-D2 shows corresponding topographical distributions of clustered electrodes. During pre- and post-onset periods (Figure 6 A1, B1), we observed strong co-activation patterns primarily among frontal and central electrodes, indicated by high-intensity diagonal blocks in co-occurrence matrices. Topographical visualization (Figure 6 A2, B2) reveals four main clusters: right frontal (blue), left frontal (orange), central (green/yellow), and posterior (red) regions. This organization reflects the initial auditory processing hierarchy with segregated functional networks responding to stimulus onset.

Pre- and post-gap periods (Figure 6 C1, D1) demonstrated reorganization of co-activation patterns with notably stronger inter-regional connectivity between frontal and parietal electrode channels, evidenced by increased off-diagonal elements in co-occurrence matrices. Topographical distributions (Figure 6 C2, D2) show cluster composition shifts, particularly in central regions where electrodes exhibit enhanced coupling with posterior sites. This suggests adaptive network reconfiguration for gap detection processing. Quantitatively, clustering analysis yielded stable clusters with average Silhouette score of approximately 0.75, indicating robust cluster

separation. Temporal evolution suggests dynamic functional network reconfiguration during task processing, with initial frontal-posterior segregation in pre-gap periods transitioning to integrated processing patterns post-gap. Cross-validation using even and odd numbered recordings separately confirmed robustness of identified electrode channel groupings, validating the biological plausibility of observed spatiotemporal patterns. Motivated by these biologically plausible clustering results demonstrating systematic reorganization of neural networks during auditory processing, we constructed a dual-channel feed-forward neural network aimed at learning latent embeddings of source-level data and predicting sensor-level activity from high-dimensional cortical representations.

### B. DualPathEEG Reconstructs Source-to-Sensor Neural Markers

Our DualPathEEG architecture effectively reconstructed sensor space EEG signals from source space representations.

TABLE I
MEAN ± STD METRICS ACROSS 5 CROSS-VALIDATION FOLDS OVER ONSET AND GAP STIMULUS. DUALPATH AE IS OUR PROPOSED ARCHITECTURE. SINGLEPATH AE IS ABLATION OF MLP LAYERS. DUALPATH GAN ADDS ADVERSARIAL TRAINING.

|  | Metric | DualPath AE | SinglePath AE | DualPath GAN |
|---|---|---|---|---|
| Train | ↓ MAE | **0.765 ± 0.286** | 0.893 ± 0.255 | 0.819 ± 0.338 |
| | ↓ MSE | **1.097 ± 1.280** | 1.546 ± 1.329 | 1.380 ± 1.877 |
| | ↓ RMSE | **0.979 ± 0.372** | 1.190 ± 0.361 | 1.060 ± 0.451 |
| | ↑ Corr. | **0.402 ± 0.253** | 0.020 ± 0.181 | 0.373 ± 0.261 |
| Val | ↓ MAE | **0.802 ± 0.208** | 0.891 ± 0.239 | 0.809 ± 0.208 |
| | ↓ MSE | **1.156 ± 0.798** | 1.524 ± 1.014 | 1.177 ± 0.827 |
| | ↓ RMSE | **1.036 ± 0.284** | 1.190 ± 0.326 | 1.045 ± 0.292 |
| | ↑ Corr. | **0.219 ± 0.283** | 0.018 ± 0.174 | 0.207 ± 0.213 |

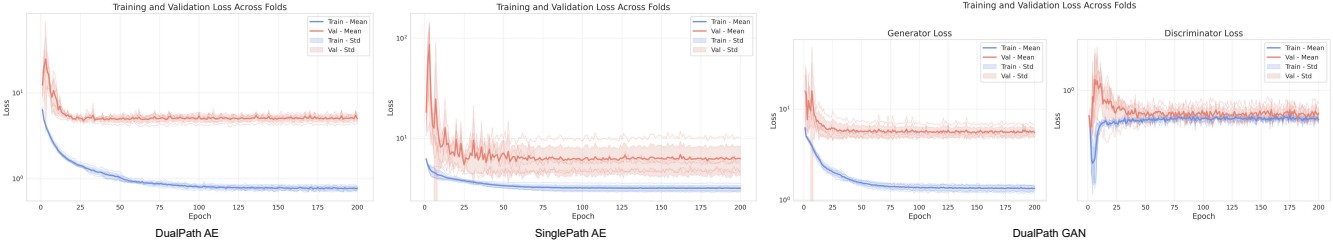

Fig. 7. DualPath AE, SinglePath AE, and DualPath GAN training and validation loss across 5 cross-val folds

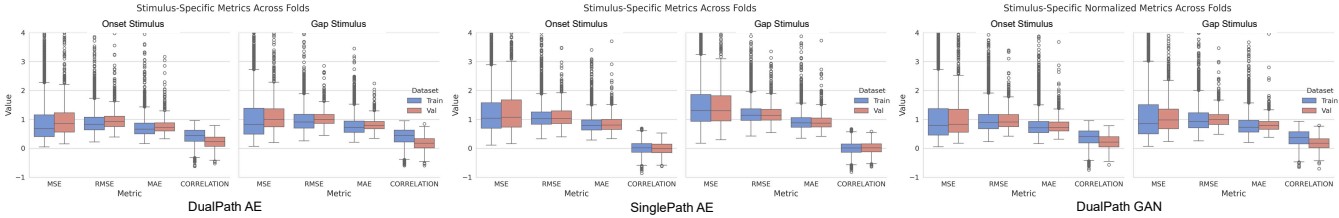

Fig. 8. DualPath AE, SinglePath AE, and DualPath GAN stimulus-specific metrics across 5 cross-val folds

We tested three variants: the proposed DualPath AutoEncoder (AE) shown in Figure 1, a SinglePath AE that ablates the initial MLP layers, and a DualPath GAN with adversarial training and a CNN discriminator. The MLP block efficiently reduces features from 20484 to 1024 through aggressive downsampling, and its removal in SinglePath AE degraded performance. The DualPath GAN uses adversarial loss to reject unrealistic signal artifacts. All models converged over 200 epochs (Figure 7), with training loss decreasing from ∼4.0 to ∼1.0 and validation loss stabilizing around epoch 100 after initial volatility. The training-validation gap indicates generalization challenges due to limited samples ($n_{train} = 797$, $n_{val} = 199$ from 38 subjects). We addressed this through 5-fold cross-validation to reduce variance and overfitting.

Performance metrics revealed significant differences between model architectures (Table I). Our proposed DualPath AE demonstrated superior performance across all metrics. During training, DualPath AE achieved MSE of 1.097 ± 1.280, substantially outperforming SinglePath AE (1.546 ± 1.329) and DualPath GAN (1.380 ± 1.877). MAE values confirmed this effectiveness: DualPath AE yielded 0.765 ± 0.286 compared to 0.893 ± 0.255 for SinglePath AE and 0.819 ± 0.338 for DualPath GAN. Validation performance remained robust with MSE of 1.156 ± 0.798 and MAE of 0.802 ± 0.208. Most notably, correlation metrics revealed significant advantages for DualPath AE, achieving 0.402 ± 0.253 (training) and 0.219 ± 0.283 (validation), substantially exceeding SinglePath AE's near-zero correlations (0.020 ± 0.181 training, 0.018 ± 0.174 validation) and outperforming DualPath GAN (0.373 ± 0.261 training, 0.207 ± 0.213 validation). Figure 8 illustrates stimulus-specific metrics across models. Unpaired t-tests across 5 cross-validation folds revealed no significant differences between training and validation performance, confirming robust generalization. These results demonstrate that the AE-based dual-pathway approach

effectively captures stimulus-specific features despite challenging dimensional reduction from 20,484 source to 32 sensor dimensions. The dedicated onset and gap MLP pathways provide crucial representational capacity lacking in single-pathway architectures. Total training carbon footprint was 0.89 kg $CO_2$eq (4.17 hours on NVIDIA RTX 4090, 475 g $CO_2$eq/kWh) [20].

Figure 9 illustrates the ground-truth and predicted temporal EEG reconstruction and their respective topographical maps for both onset and gap stimulus, revealing consistent patterns across the four selected patients. The reconstructed time-series captures the general waveform trends, and the predictions exhibited some deviations in amplitude and timing compared to ground-truth signals. Correlation coefficients provide quantitative evidence of reconstruction quality, ranging from moderate (0.4018 for patient 9C onset stimulus) to relatively strong (0.7293 for patient 9C gap stimulus). The model generally performed better with gap stimuli (also seen in figure 8), achieving consistently higher correlation values across patients. Notable challenges included difficulty reproducing sharp transitions and peak amplitudes, particularly evident in patients 9A and 9B. Peak amplitudes were often underestimated, especially in cases where the ground-truth signal showed sudden large deviations. MSE and MAE metrics further confirm varying reconstruction accuracy, with lower errors observed in patient 9A onset stimulus (MSE: 0.2364) compared to patient 9B onset stimulus (MSE: 6411). The temporal alignment between predicted and ground-truth signals varies across samples, with some predictions showing phase shifts relative to the actual measurements.

Figure 9 ground-truth and reconstructed EEG topographical maps for both onset and gap conditions shows voltage distributions represented on a scale of range -4.0 μV (blue) to +4.0 μV (red). In patient Figure 9A the ground-truth onset map shows a distributed pattern of activation spread across

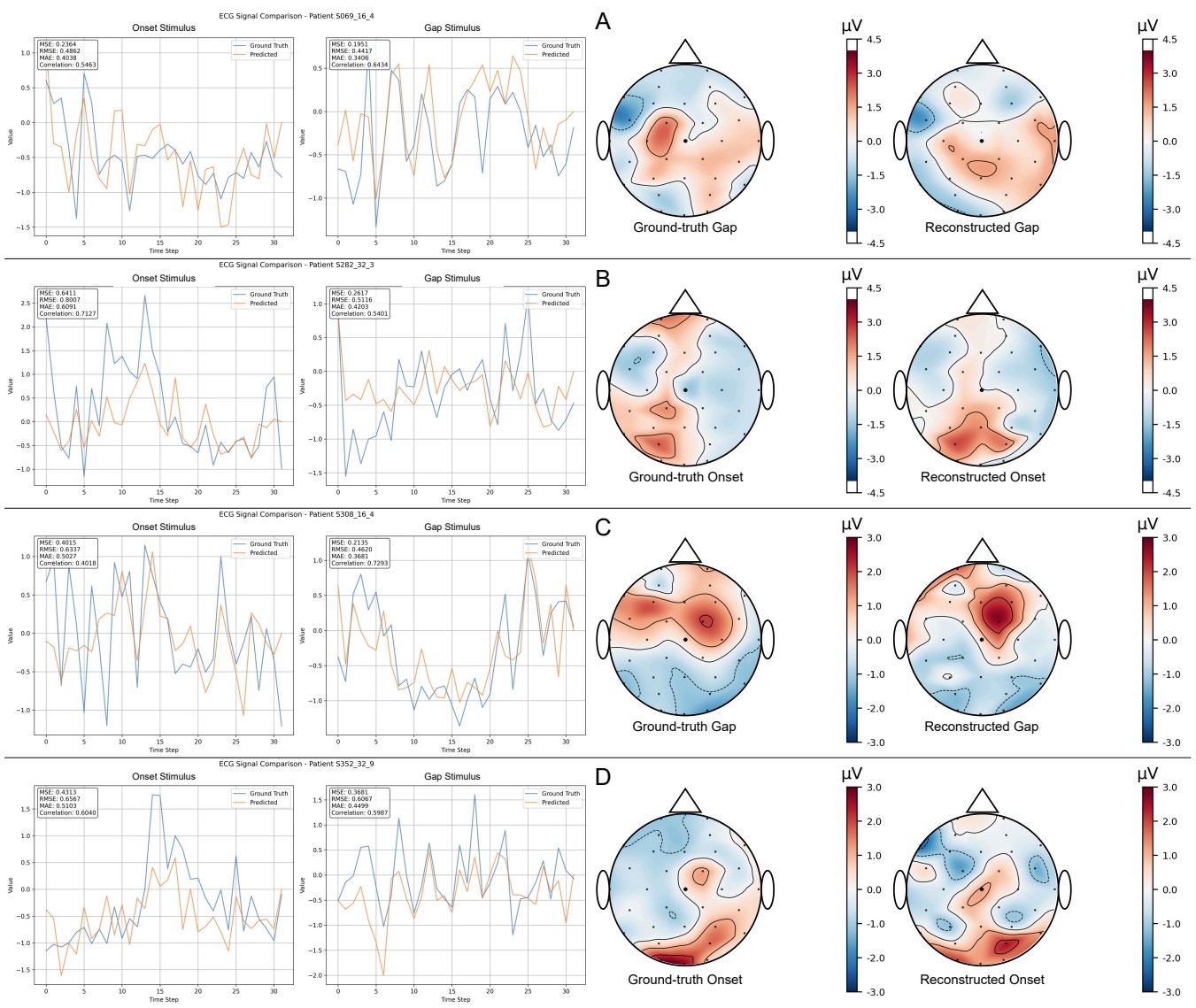

Fig. 9. DualPathEEG ground-truth vs. predicted values for the reconstructed EEG signal for both onset and gap stimulus, and its respective topographical maps. These four patient samples were selected from the best performing cross-validation fold. The inference was performed using the model from the final epoch on the validation set.

the right temporal and the central regions. The reconstructed onset demonstrates similar activity spread across with slight variation, possibly because of minor difference across ground-truth and predicted voltage values. Patient Figure 9B exhibits strong frontal parietal activations in ground-truth conditions, which are largely preserved in the reconstructed map with slightly reduced area spread in the frontal regions. Patient Figure 9C map reveals similar activation spread across the frontal and left temporal regions, which were mostly conserved in both the ground-truth and the reconstructed topological map. Similarly, in patient Figure 9D the ground-truth map and the reconstructed map show similar activation regions across parietal and occipital regions. Across all cases, our reconstruction algorithm predicted very similar activation regions with the ground-truth, showing it's capability to capture global voltage

difference across the electrodes.

## IV. DISCUSSION

We introduce two complementary deep learning approaches for EEG analysis: a biLSTM autoencoder with UMAP reduction for unsupervised functional clustering, and the Dual-PathEEG model for source-to-sensor reconstruction. The biL-STM autoencoder revealed dynamic network reconfiguration during auditory processing with robust clusters (Silhouette scores averaging 0.75), showing initial frontal-posterior segregation pre-gap transitioning to integrated processing post-gap. Lateral temporal electrodes exhibited weak co-occurrence patterns, suggesting relative independence, while midline electrodes demonstrated consistent regional clustering as potential hub nodes. These findings align with behaviorally relevant

onset and gap events engaging distributed cortical circuits including frontal salience and prediction error systems [12] [13]. The DualPathEEG architecture successfully addressed source-to-sensor transformation through dual-pathway design. Dedicated MLP processors for onset and gap stimuli proved critical, significantly outperforming single-pathway models. The CNN pathway with dilated convolutions captured multi-scale temporal dependencies, while hybrid loss ensured time-frequency domain fidelity. Results demonstrated effective dimensionality reduction and successful reconstruction of physiologically relevant features with good generalization across held-out patients. Some reconstructions showed enhanced central Cz activation compared to ground-truth, potentially from template surface model limitations and electrode placement variations. Our loss function prioritizes global accuracy, favoring high-energy central electrodes; future subject-specific models with spatial regularization should address this. Despite these limitations, polarity patterns and spatial distributions remained consistent, indicating good fidelity. Remaining challenges include reconstructing sharp transitions and peak amplitudes. Future work will incorporate anatomical information, explore sophisticated architectures, and evaluate across sensory modalities like MRI. Applications include clinical neurological diagnosis, enhanced brain-computer interfaces, and functional network identification. By combining unsupervised clustering with deep learning reconstruction, our framework bridges source-sensor space representations in neuroimaging.

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

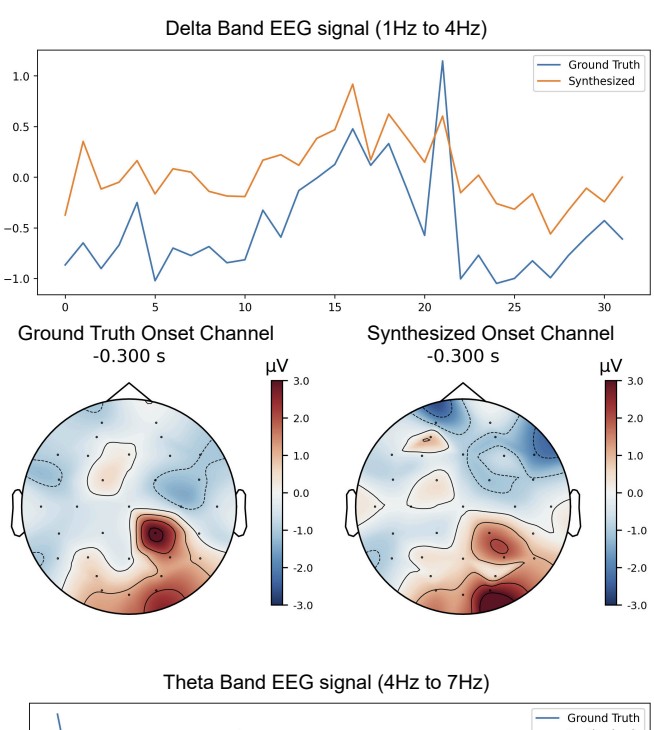

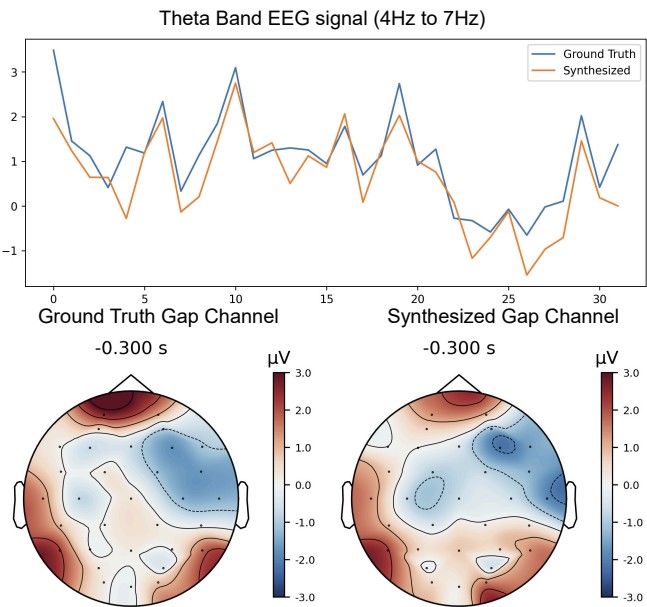

Fig. 10. DualPathEEG ground-truth vs. predicted values for the reconstructed EEG signal for both onset and gap stimulus, and its respective topographical maps for Delta (1Hz to 4Hz) and Theta (4Hz to 7Hz) bands