# OpenReview forum: "Framework for Functional Clustering and Source-to-Sensor Reconstruction of Temporally Evoked Neural Features"
_IEEE.org/EMBS/BHI/2025/Conference — BHI 2025_

### Official Review · Reviewer_RPs1 · 2025-07-08
**Readability, Spectral Validation, and Generalization Concerns**

**Confidence:** 3
**Clarity Of Writing:** fair
**Clinical Significance:** good
**Methodological Novelty:** good
**Overall Rating:** 6
**Final Rating:** 7

**Experiments And Results:**

good

**Questions For The Authors:**

1. Could you enhance the clarity of the manuscript for new readers by reducing verbosity?

2. Could you illustrate how well the model preserves signal-frequency details—for example, by reporting a simple spectral metric—and if frequency content is not critical, please discuss why.

3. Could you describe your systematic process for selecting and tuning model hyperparameters and other settings?

4. Can your approach be validated on different EEG datasets or paradigms?

**Strengths:**

1. The study showed high reconstruction accuracy that generalizes across subjects, suggesting template-based source estimates capture meaningful anatomical information.

2. Eliminates the need for individual MRI scans, greatly improving scalability and reducing cost in EEG-based research and applications.

3. Ablation studies demonstrate that DualPathEEG outperforms both single-path and adversarial variants, underscoring the critical role of its dual-path architecture for accurate source-to-sensor reconstruction.

**Summary Of The Paper:**

The study presents a two-stage, MRI-free EEG framework that first employs an unsupervised bi-LSTM autoencoder to discover spatiotemporal clusters of electrodes during an auditory tone-gap task and then uses a novel DualPathEEG dilated-CNN to reconstruct 32-channel scalp signals from template-based cortical source estimates and stimulus features. By validating those source estimates via reconstruction accuracy, the method removes the need for individual MRI scans, paving the way for more scalable, real-time EEG neuroimaging and brain-computer interface applications.

**Weaknesses:**

1. The writing is quite verbose and sometimes confusing, which may make it hard for readers—especially newcomers—to grasp the real-world applications and workflow at first glance.

2. No explicit evaluation of spectral-domain fidelity is reported—while a spectral loss is used during training, the results only show time-domain metrics (MSE, MAE, correlation), so frequency-content preservation remains unquantified.

3. The paper doesn’t describe how its many hyperparameters (e.g. loss weights (alpha and beta), cluster settings, weight decay, learning rate, etc) were chosen or tuned, so it’s unclear whether the model is optimally configured or simply hand-picked.

---

### Official Review · Reviewer_gnMG · 2025-07-15
**Framework for Functional Clustering and Source-to-Sensor Reconstruction of Temporally Evoked Neural Features**

**Confidence:** 4
**Clarity Of Writing:** good
**Clinical Significance:** good
**Methodological Novelty:** good
**Overall Rating:** 7

**Experiments And Results:**

good

**Questions For The Authors:**

Generalization Beyond Auditory Tasks: Could this framework generalize to visual/somatosensory stimuli? Have you tested other paradigms?

**Strengths:**

Novelty: The integration of unsupervised clustering with source-sensor reconstruction addresses a critical gap in EEG interpretation.
Generalization: Cross-subject validation and stable clustering demonstrate scalability beyond subject-specific models.

**Summary Of The Paper:**

The paper introduces a dual deep-learning framework for EEG analysis: (1) a BiLSTM autoencoder for unsupervised clustering of sensor-space dynamics during auditory events, and (2)  a CNN-based architecture to reconstruct sensor-space EEG from source-space data. The approach addresses limitations of traditional methods by avoiding subject-specific anatomical data. Results show robust clustering and accurate reconstruction, with cross-subject generalization.

**Weaknesses:**

Limited Biological Interpretation: Clusters (e.g., frontal-central groups) are not linked to known auditory pathways (e.g., tonotopy, superior temporal gyrus). Discussing neuroanatomical plausibility would strengthen findings.

---

### Official Review · Reviewer_3EBM · 2025-07-15
**Review of Framework for Functional Clustering and Source-to-Sensor Reconstruction of Temporally Evoked Neural Features**

**Confidence:** 3
**Clarity Of Writing:** good
**Clinical Significance:** great
**Methodological Novelty:** good
**Overall Rating:** 6
**Final Rating:** 7

**Experiments And Results:**

good

**Questions For The Authors:**

Only 38 subjects and two stimulus types; generalizability to other tasks or larger electrode montages is unclear. For improving external validity, consider testing on open datasets?

**Strengths:**

Accurate source-to-sensor modelling can improve BCIs and diagnostics without MRI-intensive head models.
Coupling unsupervised clustering with supervised reconstruction gives complementary insights into functional organization and forward mapping.
Dedicated onset/gap MLP branches and dilated CNN encoder–decoder are justified, and ablation/GAN comparisons support the architectural choices.
Co-occurrence matrices with montage overlays (Fig. 6) and ground truth vs. reconstructed topographies (Fig. 9) ease interpretation for clinicians and engineers.

**Summary Of The Paper:**

The manuscript presents a two-part deep learning framework for analysing EEG recorded during an auditory tone gap paradigm (38 adults, 32 channels). Part 1 is an unsupervised pipeline combines a bidirectional LSTM autoencoder UMAP dimensionality reduction. And Part 2 is a DualPathEEG network that reconstructs 32 channel sensor RMS responses for “onset” and “gap” events from 20 484 node source space activity. The authors argue that (a) clustering uncovers biologically plausible functional networks and (b) the reconstruction model generalizes across subjects, offering a data driven alternative to physics based forward models.

**Weaknesses:**

Cluster evaluation lacks ground truth. Silhouette ≈ 0.75 is promising but unsupervised; no behavioral or anatomical labels are used which limits biological interpretability. The authors can correlate cluster membership with auditory ERP amplitudes or cortical ROIs to validate significance.

There is a typo in the Methods section C – “pzathway”

The carbon footprint calculation of training is not supported by any reference. Provide a reference for NVIDIA RTX 4090, 475 g CO2eq/kWh.

---

### Official Review · Reviewer_Zh9X · 2025-07-17
**EEG source-to-sensor reconstruction**

**Confidence:** 3
**Clarity Of Writing:** good
**Clinical Significance:** fair
**Methodological Novelty:** fair
**Overall Rating:** 6

**Experiments And Results:**

good

**Questions For The Authors:**

- I guess F in Equation 2 is Fourier operator?
- can you get better performance by giving electrode position is input to the neural network?

**Strengths:**

- the model design is detailed and can benefit from future implementation
- the experimental design is well documented

**Summary Of The Paper:**

This paper presents a source-to-sensor reconstruction technique for EEG data. A two-part neural network design is proposed including a 1) reconstruction network DualPathEEG that predicts back projected sensor signal, and 2) a bi-directional LSTM to perform UMAP reduction and clustering. Evaluation is performed on a lab-curated dataset, and the experimental results the proposed architecture achieves strong performance compared other baseline models.

**Weaknesses:**

- lack a baseline comparison using a public dataset and existing methods
- some parameters in equation are not fully explained, along some format and typos

---

### Official Review · Reviewer_dcfx · 2025-07-21
**Framework for Functional Clustering and Source-to-Sensor Reconstruction of Temporally Evoked Neural Features**

**Confidence:** 3
**Clarity Of Writing:** good
**Clinical Significance:** fair
**Methodological Novelty:** fair
**Overall Rating:** 4

**Experiments And Results:**

good

**Questions For The Authors:**

1. Full Time‑Series Reconstruction: Can the DualPathEEG be extended to reconstruct continuous EEG waveforms (beyond RMS) and preserve phase or spectral details?

2. Baseline Comparisons: How does your method compare quantitatively (MSE, correlation) to MNE or beamformer reconstructions on the same dataset?

3. Ablation Studies: Which architectural elements (MLP vs. CNN path, dilation rates, hybrid loss weights α/β) critically impact reconstruction accuracy?

4. Generalizability Across Paradigms: Have you tested clustering and reconstruction on other sensory modalities or cognitive tasks to assess framework flexibility?

5. Anatomical Variability: Given use of a template head model, how sensitive are results to subject‑specific anatomy? Would individual MRI‑based source spaces improve clustering and reconstruction?

**Strengths:**

1. Unsupervised Functional Clustering: The biLSTM + UMAP + DBSCAN pipeline uncovers biologically plausible electrode clusters (frontal, central, posterior) and their reconfiguration between pre‑ and post‑events, offering novel insights into temporal network dynamics.

2. Dual‑Path Reconstruction Architecture: Separation into stimulus‑specific MLP paths and a dilated‑CNN pathway effectively balances aggressive dimensionality reduction with multi‑scale temporal feature capture, validated by superior reconstruction metrics across folds.

3. Cross‑Subject Generalization: Use of an fsaverage template head model and training/validation splits across subjects (80/20) with 5‑fold CV demonstrates that the framework generalizes beyond individual participants, an important step toward real‑world BCI applications.

4. Comprehensive Metrics and Visualization: Reporting of point‑wise losses, correlation coefficients, and topographical reconstructions (Fig. 9) affords a multi‑faceted evaluation of model performance, including qualitative waveform alignment and spatial map fidelity

**Summary Of The Paper:**

This manuscript presents a two‑stage deep learning framework for EEG analysis in an auditory tone‑gap paradigm. First, a bi‑directional LSTM autoencoder compresses 32‑channel, event‑locked EEG epochs into a 64‑dimensional latent space; UMAP reduction and DBSCAN clustering then reveal dynamic spatiotemporal co‑activation patterns before and after stimulus onset and gap events. Second, the DualPathEEG network reconstructs RMS‑normalized sensor‑space EEG features from high‑dimensional source‑space activations via dual MLP pathways for onset and gap events and a dilated‑CNN encoder‑decoder, trained with a hybrid time‑domain and spectral loss. Five‑fold cross‑validation demonstrates that the proposed DualPath AE outperforms both a SinglePath ablation and an adversarial GAN variant on MSE, MAE, RMSE, and Pearson correlation metrics.

**Weaknesses:**

1. Limited Reconstruction Scope: Reconstruction is confined to RMS‑normalized event‑locked features rather than full time‑series, obscuring how well transient dynamics and phase information are preserved. Only two summary metrics (RMSnorm) are modeled.

2. Absence of Classical Baselines: No quantitative comparison against traditional source‑localization methods (e.g., MNE, beamformers) on the reconstruction task, leaving unclear whether deep models offer clear advantages over well‑established inverse solutions.

3. Architectural Complexity Justification: The choice of four dilated‑CNN blocks, specific dilation rates, and UMAP hyperparameters lacks empirical ablation; it is unclear which components most drive performance.

4. Statistical Rigor and Variability Analysis: Results report mean ± std but omit statistical tests (e.g., paired t‑tests) between models or event types; subject‑level variability (e.g., worst vs. best performer) is not analyzed, potentially masking outliers.

5. Reproducibility Details: Key implementation aspects—such as DBSCAN parameters, training time per fold, code availability, and hardware specifics beyond GPU model—are not fully disclosed, hindering reproducibility.